# Microstructure, Magnetic Properties, and Application of FINEMET-Type Alloys with Co Addition

**Agnieszka Łukiewska [1], Mirosław Łukiewski [2], Mariusz Hasiak [3],\* and Hanna Łukiewska [4]**

1  Department of Physics, Częstochowa University of Technology, 42-201 Częstochowa, Poland; agnieszka.lukiewska@pcz.pl
2  TRAFECO Transformers & Inductive Components, 42-283 Boronów, Poland; m.lukiewski@trafeco.pl
3  Department of Mechanics, Materials and Biomedical Engineering, Wrocław University of Science and Technology, 50-370 Wrocław, Poland
4  Faculty of Electrical Engineering, Wrocław University of Science and Technology, 50-370 Wrocław, Poland
\*  Correspondence: mariusz.hasiak@pwr.edu.pl; Tel.: +48-713203496

**Abstract:** The choice of materials for cores of electrotechnical devices is currently related to energy saving and global warming problems. Nanocrystalline alloys are emerging as materials for cores in these devices in addition to amorphous materials already commonly used due to their better magnetic properties at high operating frequencies. The thermal stability of the magnetic properties of cores is also an important criterion. Keeping these criteria in mind, a study of microstructure and magnetic properties was carried out in this work, and FeCoNbBCu-type material was selected for use as the core of a choke operating in a DC/DC converter in interleaved topology. On the basis of the conducted studies, it was found that good magnetic properties and the best thermal stability were shown by $Fe_{58}Co_{25}Nb_3B_{13}Cu_1$ alloy. Using RALE software, the technical parameters of the choke core were determined and compared with the same parameters for a choke core made of FINEMET-type alloy.

**Keywords:** FeCo-based nanocrystalline alloys; microstructure; magnetic properties; magnetic cores

## 1. Introduction

Due to the demand for smaller and more efficient devices, nanocrystalline soft magnetic materials are increasingly used in electronic, power electronic, telecommunication, automotive, and railway systems [1]. These applications place high requirements on magnetic cores in terms of mechanical properties and efficient operation over a wide range of frequencies and temperatures. Geometric size, weight, and, above all, the price of the equipment are also important for the customer. In order to achieve these objectives, core materials must exhibit high saturation induction and magnetic permeability. Furthermore, the coercive field and hysteresis and eddy current losses must be as low as possible [2–4]. Over the past few decades, nanocrystalline materials have been obtained that more or less meet these requirements. Among the commercially available nanocrystalline materials, the most popular ones are FINEMET- (Fe-Si-B-Nb-Cu), NANOPERM- (Fe-M-B-Cu, M = Zr, Nb, Hf), and HITPERM (FeCo-M-B-Cu, M = Zr, Nb, Hf)-type alloys [5].

Nanocrystalline Fe-based alloys are usually obtained by the controlled crystallization of amorphous precursors that are manufactured by various methods in the form of powders [4,6,7], wires [4,8,9], or thin ribbons [4,5]. While wires are used in microelectronics, spintronics, or as sensors [8,9], ribbons or powders are mainly used for the cores of power equipment [10,11].

The microstructure and magnetic properties of nanocrystalline alloys strongly depend on the chemical composition and the heat treatment conditions of Fe-based amorphous alloys [12–15]. This makes it easy to form the properties of nanocrystalline alloys for different applications.

Nanocrystalline FINEMET-type alloys show excellent soft magnetic properties at room temperature and low-frequency conditions up to 20 kHz. This is related to their microstructure and depends on the exchange coupling between $\alpha$-FeSi grains through amorphous boundaries [16]. This interaction significantly weakens with increasing temperature, and it is finally suppressed when the temperature approaches the Curie temperature of the amorphous matrix of these alloys [17]. For higher frequencies, a rapid increase in core losses is observed [18]. This behavior limits the use of this type of material for the cores of devices operating at elevated temperatures and high frequencies.

HITPERM and NANOPERM alloys behave much better under these conditions, but they are not used in the industry due to their high price because of the presence of elements such as Zr, Hf, and Nb in their composition. Moreover, the presence of these elements not only increases the material cost, but also results in a major decrease in saturation induction. Therefore, other ways to improve the magnetic properties of core materials have been sought, especially under extreme operating conditions. It is known that the stability of the soft magnetic properties towards high temperatures is significantly improved with an addition of cobalt [19] because the Co atoms enhance the Curie temperature to about 1100 K depending on the chemical composition of the alloys [20]. A lot of research results have shown that the partial substitution of Co for Fe is an effective way to improve high-temperature and high-frequency characteristics [5,21,22]. Therefore, in this paper, the authors propose to study FINEMET-based material in which some Fe atoms are replaced by Co atoms. Optimal properties have been found for alloys with equal amounts of Fe and Co atoms [23].

Due to the high price of cobalt, in this work, we used FeCoNbBCu materials with a reduced Co content for the choke core in order to reduce the cost of the device. For selected nanocrystalline $Fe_{83-x}Co_xNb_3B_{13}Cu_1$ (x = 6, 25, 58 at.%) materials, microstructure and magnetic properties measurements were carried out. Using values such as maximum induction and core losses, basic parameters for the choke core were calculated using RALE design software. In order to obtain the most stable inductance characteristics over a wide range of variations in rated current, tested materials in the form of thin ribbons were used to build multigap choke cores.

## 2. Materials and Methods

Amorphous $Fe_{83-x}Co_xNb_3B_{13}Cu_1$ (x = 6, 25, 58 at.%) alloys were prepared in a 10 mm wide and 25 $\mu$m thick ribbon form using rapid solidification on a cooper quenching wheel. To obtain the nanocrystalline structure, samples were subjected to isothermal annealing for 10 min at 750 K in a vacuum of $1.33 \times 10^{-3}$ Pa. The annealing temperature was determined by differential scanning calorymetry (DSC) using NETSCH STA 449 F1 Jupiter (NETZSCH Analysing & Testing, Ahlden, Germany) set at a heating rate of 20 K/min.

The microstructures of the as-quenched ribbons and nanocrystalline samples were examined by X-ray diffractometry, Mössbauer spectroscopy, and transmission electron microscopy. A Bruker-AXS, type D8 Advanced, X-ray diffractometer with CoK$\alpha$1 radiation (1.78897 Å) was used. Transmission Mössbauer spectra were recorded at room temperature using a conventional constant acceleration spectrometer with a $^{57}$Co(Rh) source of 50 mCi radioactivity. The spectra were analyzed using a Normos package (written by R. A. Brand) according to the procedure developed in [24]. Observations using a classical electron and high-resolution microscope, type JEM 3010, were carried out for samples in the form of discs with a diameter of 3 mm cut from strips and ion-polished to a thickness of about 100 nm.

Magnetic properties were studied using a hysteresisgraph (AMH-50K-S, Laboratorio ElletroFisico Engineering, Nerviano, Italy) for core samples with an outer diameter of 20 mm and an inner diameter of 15 mm fabricated by winding the ribbons into toroidal cores. Using the results obtained, such as maximum induction and core losses, the basic parameters for the choke cores were calculated using RALE design software (RALE RDS, version 16.81).

## 3. Results and Discussion

The investigated $Fe_{83-x}Co_xNb_3B_{13}Cu_1$ (x = 6, 25, 58) alloys in the as-quenched state were fully amorphous, which was confirmed by X-ray diffraction and Mössbauer spectroscopy measurements. X-ray diffraction patterns of the as-cast alloys show only broad bumps [25]. For nanocrystalline alloys obtained by annealing at 750 K for 10 min, sharp peaks corresponding to the crystalline phase are detected in X-ray images (Figure 1).

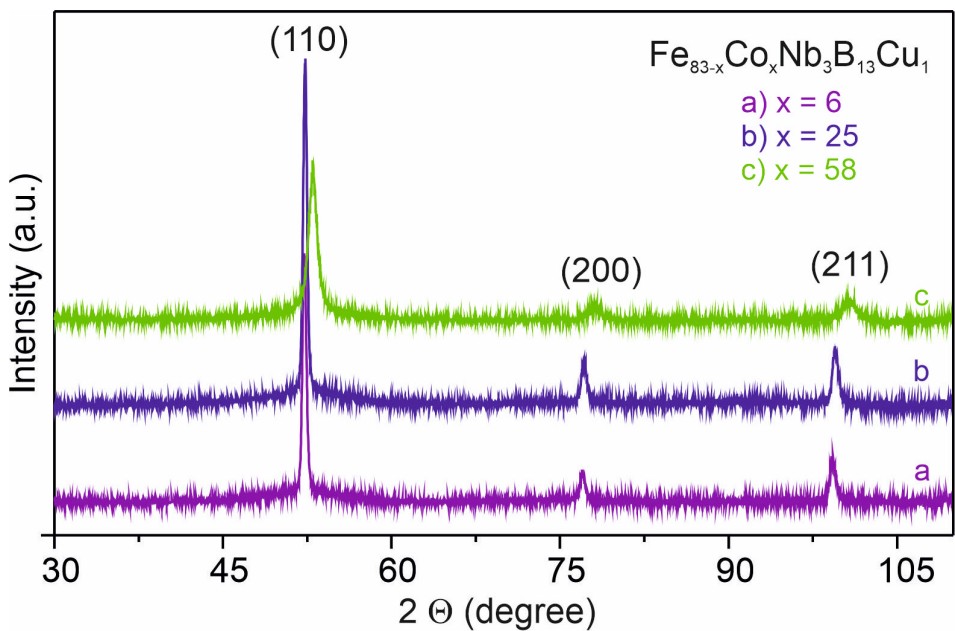

**Figure 1.** XRD patterns for $Fe_{83-x}Co_xNb_3B_{13}Cu_1$ (x = 6 (a), 25 (b), and 58 (c)) alloys annealed at 750 K for 10 min.

Amorphicity of investigated materials in the as-quenched state was also tested by Mössbauer spectroscopy. Mössbauer spectra for these samples consist of sextets with very broad and overlapped lines typical of an amorphous state [26]. Transmission Mössbauer spectra obtained at room temperature for nanocrystalline alloys are presented in Figure 2. The sextets with sharp lines related to α-FeCo crystalline grains superimposed on the broad subspectra corresponding to the remaining intergranular amorphous phase are visible. In the hyperfine magnetic field distributions P(B) of these samples, the appearance of additional high field components was assigned to this crystalline phase. The crystalline α-FeCo phase in nanocrystalline HITPERM-type alloys obtained by the conventional annealing of amorphous ribbons exhibits a long-range order of atoms [14]. However, its composition depends on the Co concentration in amorphous precursors and annealing conditions. The presence of the crystalline phase in alloys annealed at 750 K for 10 min was also confirmed by TEM images. For example, in Figure 3, micrographs and the corresponding electron diffraction pattern (an insert) of $Fe_{58}Co_{25}Nb_3B_{13}Cu_1$ alloy are shown. The presence of crystalline grains of about 10–30 nm in diameter embedded in the amorphous matrix can be observed. These grains are distributed irregularly, and in some regions, conglomerates of grains separated by the amorphous matrix are observed. The electron diffraction pattern, in addition to the broad rings that come from the amorphous phase, contains very bright reflexes, with the distances from the center of the pattern being related to the reciprocal of the interplane distances in α-FeCo phase.

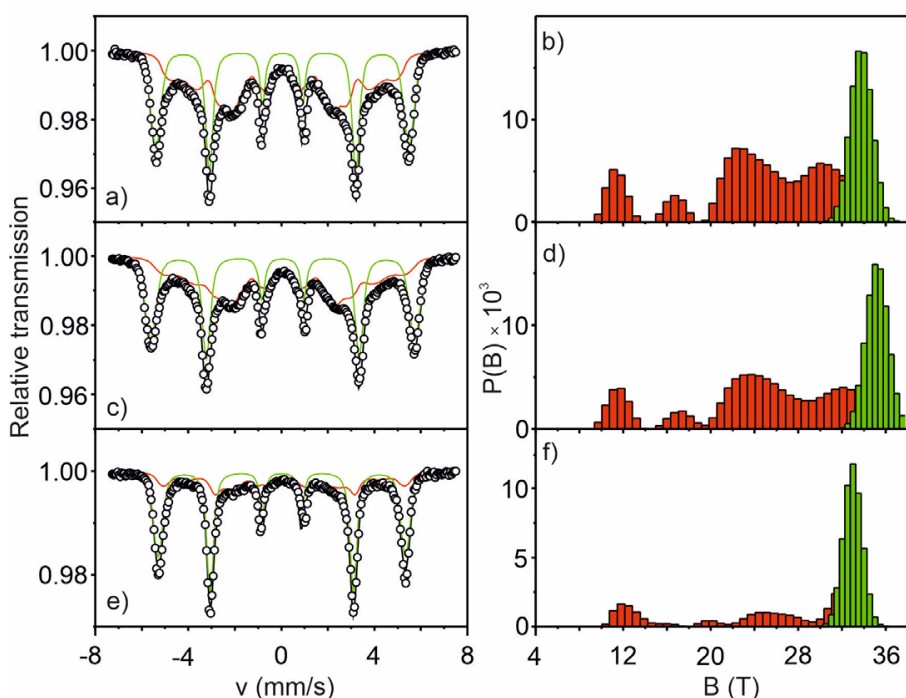

**Figure 2.** Mössbauer spectra (**a**,**c**,**e**) and hyperfine field distributions (**b**,**d**,**f**) of $Fe_{83-x}Co_xNb_3B_{13}Cu_1$ (x = 6 (**a**,**b**), 25 (**c**,**d**), and 58 (**e**,**f**)) alloys annealed at 750 K for 10 min. Reprinted and adapted from [26] with permission from Elsevier.

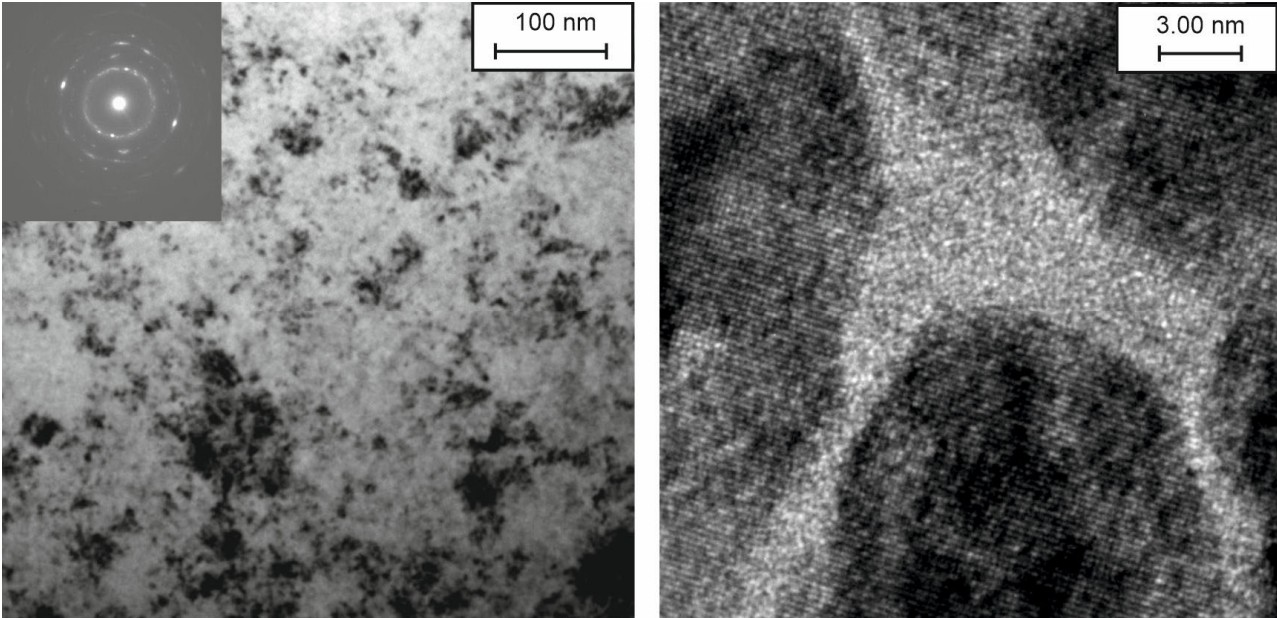

**Figure 3.** Transmission electron microscopy images and corresponding electron diffraction pattern of $Fe_{58}Co_{25}Nb_3B_{13}Cu_1$ alloy annealed at 750 K for 10 min.

The microstructure, chemical composition of the amorphous alloys, and the heat treatment conditions of these samples have a strong influence on the crystalline phase composition and magnetic properties of nanocrystalline alloys obtained by appropriate annealing from an amorphous precursor [5]. The high-value saturation magnetic flux density ($B_s$) and low coercive field ($H_c$) of soft magnetic materials are necessary for increasing the power density of modern magnetic devices. In Figure 4, hysteresis loops recorded for the investigated alloys with 6, 25, and 58 at.% Co contents (a, b, and c, respectively) are

presented. The shape of these loops is typical for soft ferromagnetic materials and indicates a low value of the coercive field of less than 125 A/m. The characteristic parameters of the tested materials determined from B-H loops (Figure 4) are shown in Table 1. It can be seen that $H_c$, as well as Bs, varies with the change in Co content. However, in the studied materials annealed at 750 K for 10 min, no relationship between the saturation magnetic flux density $B_s$ and the coercive field was observed, which presents a general trade-off in FINEMET soft magnetic materials [27].

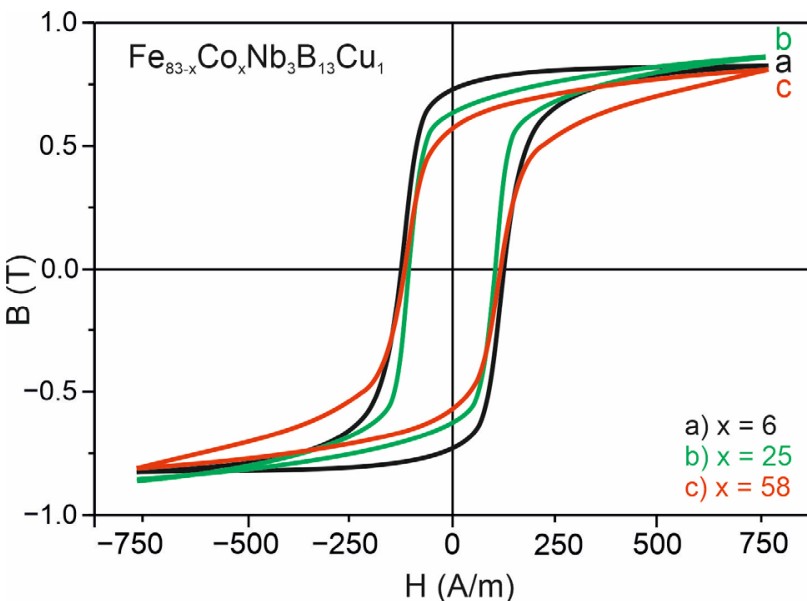

**Figure 4.** Hysteresis loops for $Fe_{83-x}Co_xNb_3B_{13}Cu_1$ (x = 6 (a), 25 (b), and 58 (c)) alloys annealed at 750 K for 10 min.

**Table 1.** Magnetic properties of $Fe_{83-x}Co_xNb_3B_{13}Cu_1$ (x = 6, 25, and 58) alloys annealed at 750 K for 10 min.

| Parameters | $Fe_{77}Co_6Nb_3B_{13}Cu_1$ | $Fe_{58}Co_{25}Nb_3B_{13}Cu_1$ | $Fe_{15}Co_{58}Nb_3B_{13}Cu_1$ |
|---|---|---|---|
| Coercive field, $H_c$ (A/m) | $120 \pm 5$ | $110 \pm 4$ | $115 \pm 5$ |
| Saturation magnetic flux density, $B_s$ (T) | $0.84 \pm 0.02$ | $0.87 \pm 0.02$ | $0.82 \pm 0.02$ |
| Remanence, $B_r$ (T) | $0.73 \pm 0.02$ | $0.68 \pm 0.02$ | $0.57 \pm 0.02$ |

Figure 5 shows the dependence of core losses on maximum induction. For comparison, the $P(B_{max})$ curve (a) for the $Fe_{83}Nb_3B_{13}Cu_1$ alloy annealed under the same heat treatment conditions was added. The studied alloys do not show large differences in the magnitude of losses, especially in the induction range up to 0.5 T. Above this induction value, the smallest losses were obtained for $Fe_{58}Co_{25}Nb_3B_{13}Cu_1$ alloy. It is worth noting that nanocrystalline Fe-Co-Nb-B-Cu alloys show good thermal stability of the magnetic properties. As can be seen in Figure 6, the core losses do not change for all of the investigated alloys at temperatures up to 500 K. Above this temperature, the losses increase sharply for the $Fe_{83}Nb_3B_{13}Cu_1$ alloy and slightly for the $Fe_{77}Co_6Nb_3B_{13}Cu_1$ and $Fe_{25}Co_{58}Nb_3B_{13}Cu_1$ alloys. $Fe_{58}Co_{25}Nb_3B_{13}Cu_1$ alloy shows the lowest core losses and the best thermal stability of core losses. The stability of magnetic properties at high temperatures is required in devices operating at an H-class temperature.

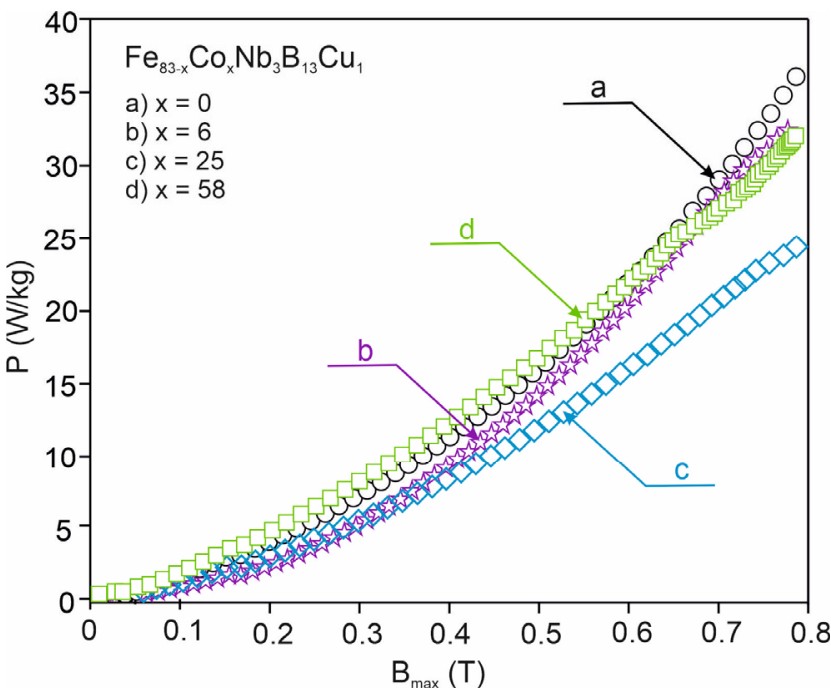

**Figure 5.** Core losses of $Fe_{83-x}Co_xNb_3B_{13}Cu_1$ (x = 0 (a), 6 (b), 25 (c), and 58 (d)) alloys annealed at 750 K for 10 min measured for the 1 kHz frequency.

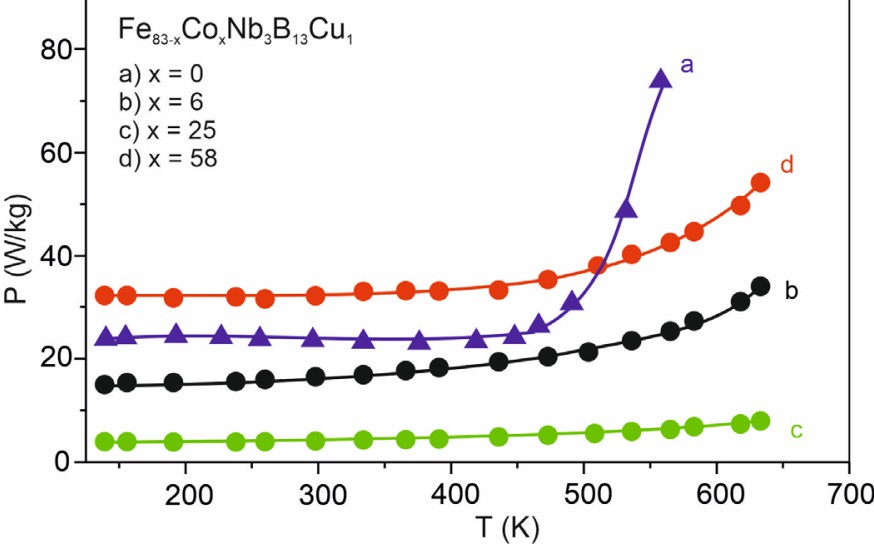

**Figure 6.** Core losses (P) as a function of temperature for $Fe_{83-x}Co_xNb_3B_{13}Cu_1$ (x = 0 (a), 6 (b), 25 (c), and 58 (d)) alloys annealed at 750 K for 10 min.

Figure 7 shows a schematic of a DC/DC converter in the interleaved topology used in the power supply systems of the rail vehicle. Three single-phase chokes operate in the output filter of the converter. The converter operates in a sealed enclosure on the roof of the rail vehicle. For this reason, reducing the weight and loss of inductive components are important design criteria. Choke designs were made for three different core materials. A well-known $Fe_{83}Nb_3B_{13}Cu_1$ alloy was taken as the base material. Then, comparative designs were made for alloys with different cobalt contents. The design input assumptions are shown in Table 2, and they characterize the electrical forcing at the choke operation point and their operating conditions.

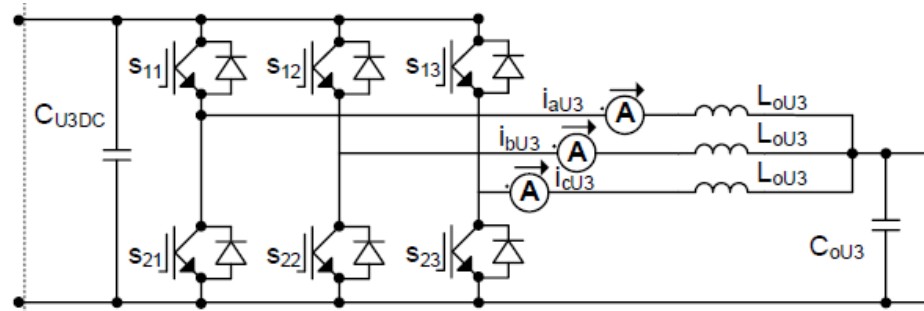

**Figure 7.** Electrical diagram of a DC/DC converter in the interleaved topology. Explanation for variables: CU3DC—DC circuit capacity, CoU3—output filter capacity, LoU3—phase inductances of filter chokes, iau3, ibu3, icu3—choke phase currents, S11–S13 and S21–S23—transistor switches.

**Table 2.** Technical specification of parameters and operating conditions of chokes.

| Input Data of the Designed Choke | |
| --- | --- |
| Nominal voltage | 900 V DC |
| Insulation voltage against earth | 4 kV DC |
| Inductivity | 500 μH |
| Nominal current | 100 A DC |
| Magnetic linearity/overload | 150 A DC |
| Ripple Current | 8 kHz—25 Arms |
| | 16 kHz—0.7 Arms |
| | 24 kHz—2.6 Arms |
| | 32 kHz—0.7 Arms |
| | 40 kHz—1.0 Arms |
| Duty cycle | S1—continuous |
| Cooling | AN—air, natural |
| Max ambient temperature | 45 °C |
| Insulations class | Class F ($T_{max}$ = 155 °C) |
| Noise level | <50 dB |

The chokes shown in Figure 7 are the output filter of the DC/DC converter. The output current of the converter flowing through the chokes contains mid-frequency current harmonics in the 8–40 kHz range, in addition to the main DC component. The dominant one is the 8 kHz component, which has the highest amplitude and causes the highest losses in the choke core. Amplitudes and frequencies of the current harmonics have a decisive influence on the value of choke losses.

The designed chokes are based on a UI-type core in block design with additional air gaps (Figure 8). Air gaps introduced in the columns of the core make it possible to achieve adequate inductance and linearity of the inductance characteristics during overloads throughout the design and subsequent calibration of the choke. Efforts have been made to minimize the width of the introduced slots due to the intense losses associated with flux dissipation around the slots [28]. A CoreECO[TM] bonded multigap choke core was designed [29]. The core would have eight slots in each column. The slots would be filled with glass-epoxy spacers. The stiffness of the core structure would be ensured by flat aluminum column bars placed on the face of the core column. The column flat bars, slotted spacers, and core blocks would be joined with high-temperature-resistant epoxy adhesive. The fill factor of the nanocrystalline iron core is assumed to be 85%.

Basic criteria for the selection of the magnetic material of the choke core are the lowest possible loss and a relatively high saturation induction (Table 3). Inductive components in power electronics are designed in temperature classes in which the choke insulation materials used can operate permanently. These are usually class F ($T_{max}$ = 155 °C) and class H ($T_{max}$ = 180 °C). The temperatures listed are the long-term permissible temperatures during normal operation of the choke under rated conditions. During overloads, higher

operating temperatures are permissible, but they are short in duration. For this reason, it is extremely important that the magnetic core maintains the temperature stability of parameters over a wide temperature range. The temperature stability of losses of the core material is particularly important. Figure 6 shows the temperature dependence of losses of the tested magnetic alloys in the temperature range up to 650 K. The best temperature stability of losses is shown for alloys with 25 % and 6 % cobalt contents.

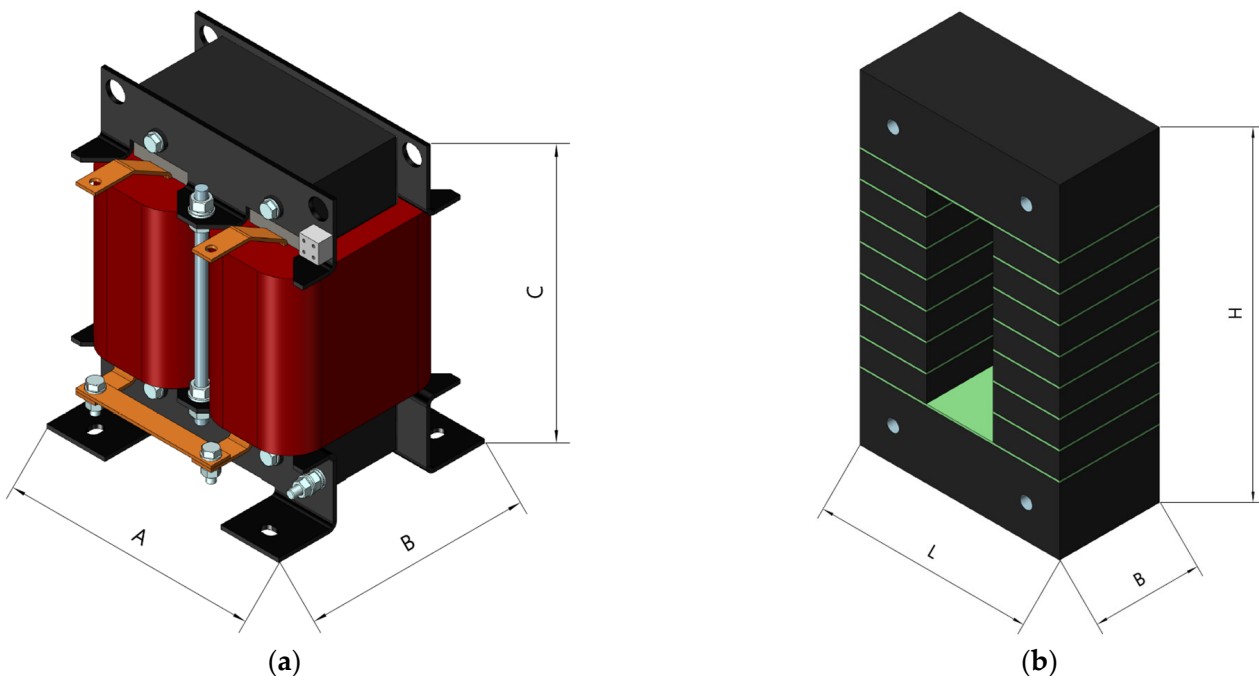

(a)　　　　　　　　　　　　　　　　　　　　　　　　　　　(b)

**Figure 8.** Single phase 2RTF type choke with nanocrystalline block core made with CoreECO multi-gap technology: dimensional draft of the choke (**a**); dimensional draft of the core (**b**). A and L, B, C and H—length, depth and height of the chokes, respectively.

**Table 3.** Choke cores magnetic materials parameters.

| Core Materials | $Fe_{83}Nb_3B_{13}Cu_1$ | $Fe_{77}Co_6Nb_3B_{13}Cu_1$ | $Fe_{58}Co_{25}Nb_3B_{13}Cu_1$ |
|---|---|---|---|
| Losses *, P (W/kg) | $21.9 \pm 0.1$ | $20.8 \pm 0.1$ | $16.3 \pm 0.1$ |
| Saturation magnetic flux density $B_s$ (T) | $1.21 \pm 0.02$ | $0.84 \pm 0.02$ | $0.87 \pm 0.02$ |

* Determined by the transformer method for a frequency of 1 kHz.

Chokes operating at medium frequencies can be a source of nuisance acoustic fields, which are mainly due to magnetostriction effects. The noise level generated by the choke is also one of the criteria for selecting the magnetic core material. In rail converters, the noise level must be particularly limited for the sake of passengers' comfort. Although some frequencies are close to the limits of human hearing, they can, nonetheless, cause annoyance. This noise is also audible to animals and can cause them additional stress and irritation. Nanocrystalline materials with cobalt content have very small magnetostriction coefficients, which are close to zero [4].

The cores of chokes are made with multi-gap technology. The use of multiple narrow gaps in the core helps to reduce dispersion losses. The material for the cores is a thin film with a thickness of about 25 μm of nanocrystalline material with a suitable alloy composition. Nanocrystalline tape is bonded with high-temperature epoxy in blocks from which the core is built [29]. Designed chokes with a column-type core have two independent winding coils connected in series. Since the main component of the choke current is the DC component, choke windings are designed from a thin aluminum sheet.

This solution makes it possible to simultaneously achieve optimal current density in the winding, reduce eddy current losses, and keep the choke mass low. Reducing the weight of chokes in rail converters that operate in rail vehicles is an important design requirement. Windings use flexible insulation with a thickness of 0.13 mm in temperature class H. Choke components are made of a paramagnetic sheet or glass-epoxy composite. Chokes made of $Fe_{77}Co_6Nb_3B_{13}Cu_1$ and $Fe_{58}Co_{25}Nb_3B_{13}Cu_1$ alloys were designed with lower induction in the core compared to that of the base material. This is due to the need to realize the condition of magnetic linearity under overload and lower saturation induction of the material. Losses in the core for the $Fe_{77}Co_6Nb_3B_{13}Cu_1$ choke are slightly lower compared to those of the base material despite the fact that the loss of magnetic material is lower by about 5%. $Fe_{58}Co_{25}Nb_3B_{13}Cu_1$ alloy has a 25% lower loss compared to that of the base material. However, the total loss of a choke based on an $Fe_{58}Co_{25}Nb_3B_{13}Cu_1$ core is reduced by only about 7% (Table 4). The disproportionate and lower-than-expected core loss reduction is caused by the dispersion of magnetic induction in the air gaps of the choke core. The magnetic flux concentrated in the core near the air gaps and in the gaps of the core itself changes its direction of path and is dispersed. Through this, it penetrates the core sheets in unfavorable directions, causing excessive eddy current losses.

**Table 4.** Technical data of chokes obtained in computational simulations.

| Technical Parameters * | Reactor Type 2RTF—50 µH 100Adc 900Vdc T40F | | |
|---|---|---|---|
| Core material | $Fe_{83}Nb_3B_{13}Cu_1$ | $Fe_{77}Co_6Nb_3B_{13}Cu_1$ | $Fe_{58}Co_{25}Nb_3B_{13}Cu_1$ |
| Core weight | | 17.8 kg | |
| Flux density | 0.52 T | 0.48 T | 0.48 T |
| Core losses | 168 W | 166 W | 132 W |
| Core temperature rise | 73 K | 72 K | 67 K |
| Winding material | aluminum sheet 3× (0.2 × 125 mm) | | |
| Winding losses | 109 W | 128 W | 126 W |
| Winding temperature rise | 69 K | 70 K | 68 K |
| Number of turns | 30 turns | 32 turns | |
| Air gaps in the core per column | 8 × 0.55 mm | 8 × 0.65 mm | |
| Core dimensions L × B × H | 180 × 90 × 260 mm | | |
| Reactor weight | 20.2 kg | 20.4 kg | |
| Reactor dimensions A × B × C | 240 × 180 × 265 mm | | |

\* Choke parameters were calculated using RALE Design System software.

A reduction by approximately 7% in choke losses and a smaller increase in the core and winding temperatures have a positive effect on the cooling of the sealed inverter housing and on the efficiency of the entire power system.

## 4. Conclusions

Nanocrystalline $Fe_{83-x}Co_xNb_3B_{13}Cu_1$ (x = 6, 25, 58) alloys, in which some Fe atoms have been replaced with Co atoms and annealed at 750 K for 10 min, can be used successfully as materials for magnetic cores. The investigated materials exhibit excellent soft magnetic properties (i.e., a coercive field of less than 125 A/m, and a saturation magnetic flux density ranging from 0.82 to 0.87 T), which are related to their microstructure, and depend on the exchange coupling between the nanocrystalline α-FeCo grains of 10–30 nm in diameter embedded in the residual amorphous matrix. It is worth noting that the investigated $Fe_{83-x}Co_xNb_3B_{13}Cu_1$ (x = 6, 25, 58) alloys annealed at 750 K for 10 min exhibit very low core losses, which makes them extremely attractive for use in the cores of electromagnetic devices. The shift of the Curie points toward higher temperatures after replacing Fe atoms with Co atoms in FINEMET-type alloys, and this makes them suitable for use in the cores of devices operating at H-class temperatures. Taking into account

the results of measurements and simulations of choke core parameters, the manufactured device uses a core made of $Fe_{58}Co_{25}Nb_3B_{13}Cu_1$ alloy. Choke specifications obtained in computational simulations with RALE software using the innovative CoreECO$^{TM}$ bonded multigap choke core system demonstrated the wide applicability of Fe-Co-Nb-B-Cu alloy in DC/DC converters.

**Author Contributions:** M.H.: conceptualization, methodology, validation, writing—original draft, writing—review and editing. M.Ł.: investigation, methodology, data curation, formal analysis. A.Ł.: conceptualization, investigation, methodology, writing—original draft, data curation, formal analysis, visualization. H.Ł.: data curation, visualization. All authors have read and agreed to the published version of the manuscript.

**Funding:** This research received no external funding.

**Conflicts of Interest:** The authors declare no conflict of interest.

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
