# Peer review of "Microstructure, Magnetic Properties, and Application of FINEMET-Type Alloys with Co Addition"

_applsci, doi:10.3390/app13084693_

Round 1
Reviewer 1 Report
The manuscript is a valuable research report on FINEMET alloys for the cores of electrotechnical devices. The research results for use of these materials as the core of a choke operating in a DC/DC converter have been discussed. In particular, Fig.6 shows convincing data on the thermal stability of their magnetic properties.
Can the authors justify shortly the choice of the annealing temperature T=750 K and time of 10 minutes? After addressing this remark, the work may be accepted for publication.
Minor language corrections are required, e.g.:
page 3, line 105 Amorphycity -> Amorphicity
page 3, line 119: ”One can be seen the crystalline grains” ?
Author Response
Dear Reviewer,
Thank you very much for all the comments and suggestions on our article. Below I am sending a list of the changes made in the article. In addition, all changes incorporating the reviewers' suggestions are highlighted in red.
Reviewer #1: Can the authors justify shortly the choice of the annealing temperature T=750 K and time of 10 minutes?
The annealing temperature of 750 K was chosen based on DSC studies. This temperature for all the investigated alloys is between the primary and secondary crystallization temperature. The annealing time of 10 min. was chosen on the assumption that the heat treatment would produce a nanocrystalline structure with a low crystalline phase content.
Reviewer #1: page 3, line 105 Amorphycity -> Amorphicity
Corrected.
Reviewer #1: page 3, line 119: ”One can be seen the crystalline grains”
This sentence was rewritten.

Reviewer 2 Report
The current work illustrate the microstructure, magnetic properties, and Application of Finemet-type Alloys with Co Addition. I found the work is quite interesting and can be published in Applied science after fixed these necessary comments;
1- The introduction is very poor and need to enhanced with more relevant references and describe with more details the different forms of these alloys (microwires, thin films, nanoparticles,…… ) and its applications and why they choose to study the alloy with ribbon form.
2- If possible, the authors should add the EDX analysis to proof the chemical composition.
3- The XRD analysis reported in figure 1 and its discussion it is very poor. The authors must add the miller parameters and estimation of the average grain size must be include in the final version of the manuscript the authors can used the similar analysis reported in this reference; (Salaheldeen, M.; Garcia, A.; Corte-Leon, P.; Ipatov, M.; Zhukova, V.; Zhukov, A. Unveiling the Effect of Annealing on Magnetic Properties of Nanocrystalline Half-Metallic Heusler Co2FeSi Alloy Glass-Coated Microwires. J. Mater. Res. Technol. 2022, 20, 4161–4172.)
4- TEM analysis need to be enhanced with low scale images to see the planers and the nancrystalline structure of the sample and it will be great if it combine with the EDX and the element distribution to have a clear idea about the homogeny of the microstructure investigation.
5- The M-H loops in figure 4 need to be remeasured as it seems that M-H loops of (b, x = 25) and (c x = 58) are unsaturated such as the hysteresis loop of (a). In addition, the main estimate magnetic parameters such as the coercivity Hc, Ms and Mr must be include in the revised version of the manuscript to have a better understanding of the magnetic behavior.
6- The authors should add the uncertainties others measurements at table 1, 2 and 3.
7- The conclusion need to enhance with more data about the microstructure and structure outcome and define the possible applications.
8- The reference must be updated with more recent work from other research group.
Author Response
Dear Reviewer,
Thank you very much for all the comments and suggestions on our article. Below I am sending a list of the changes made in the article. In addition, all changes incorporating the reviewers' suggestions are highlighted in red.
Reviewer #2: The introduction is very poor and need to enhanced with more relevant references and describe with more details the different forms of these alloys (microwires, thin films, nanoparticles,…… ) and its applications and why they choose to study the alloy with ribbon form.
The introduction was rewritten according to the reviewer’s suggestions.
Reviewer #2: If possible, the authors should add the EDX analysis to proof the chemical composition.
The chemical composition of the studied alloys was not studied by SEM/EDX. The reason for this was the nanocrystalline grain size embedded in the amorphous matrix, which can not be seen in these observations (due to the SEM resolution). Therefore, it was assumed that the chemical composition of the alloy corresponds to the content of the components used to make the alloys, especially since no mass loss (less than 0.25%) was found in the process of making the ingots.
Reviewer #2: The XRD analysis reported in figure 1 and its discussion it is very poor. The authors must add the miller parameters and estimation of the average grain size must be include in the final version of the manuscript the authors can used the similar analysis reported in this reference; (Salaheldeen, M.; Garcia, A.; Corte-Leon, P.; Ipatov, M.; Zhukova, V.; Zhukov, A. Unveiling the Effect of Annealing on Magnetic Properties of Nanocrystalline Half-Metallic Heusler Co2FeSi Alloy Glass-Coated Microwires. J. Mater. Res. Technol. 2022, 20, 4161–4172.)
The Miller parameters on XRD patterns were added to Fig. 1. The average grain size of a-FeCo phase estimated from TEM analysis was added to manuscript. Moreover, the additional transmission electron microscopy image with 3 nm scale for the Fe58Co25Nb3B13Cu1 alloy annealed at 750 K for 10 min. was added to the manuscript as Fig. 3.
Reviewer #2: TEM analysis need to be enhanced with low scale images to see the planers and the nancrystalline structure of the sample and it will be great if it combine with the EDX and the element distribution to have a clear idea about the homogeny of the microstructure investigation.
The additional TEM image with 3 nm scale was added to Fig. 3, righ side.
Reviewer #2: The M-H loops in figure 4 need to be remeasured as it seems that M-H loops of (b, x = 25) and (c x = 58) are unsaturated such as the hysteresis loop of (a). In addition, the main estimate magnetic parameters such as the coercivity Hc, Ms and Mr must be include in the revised version of the manuscript to have a better understanding of the magnetic behavior.
The B-H loops recorded at room temperature were remeasured and added to the manuscript according to reviewer’s suggestion. Moreover, the magnetic parameters were evaluated and additional listed in Table 1.
Reviewer #2: The authors should add the uncertainties others measurements at table 1, 2 and 3.
Uncertainties in the measured physical quantities were added to the manuscript. Only Input data to the designed choke as well as technical data of chokes obtained in computational simulations (RALE design software) are not expressed with errors due to the nature of the design process.
Reviewer #2: The conclusion need to enhance with more data about the microstructure and structure outcome and define the possible applications.
The summary of microstructure studies was rewritten and enriched with data obtained in measurements.
Reviewer #2: The reference must be updated with more recent work from other research group.
As suggested by the reviewer, additional papers were added to the manuscript as references to update our paper.

Round 2
Reviewer 2 Report
The authors fixed most of my comments. I feel the current version of the manuscript is suitable for publish